# The Mechanism of Bacterial Resistance and Potential Bacteriostatic Strategies

**DOI:** 10.3390/antibiotics11091215

**Published:** 2022-09-08

**Authors:** Fusheng Zhang, Wei Cheng

**Affiliations:** Division of Respiratory and Critical Care Medicine, Respiratory Infection and Intervention Laboratory of Frontiers Science Center for Disease-related Molecular Network, State Key Laboratory of Biotherapy, West China Hospital of Sichuan University, Chengdu 610041, China

**Keywords:** bacterial drug resistance, new antibacterial compounds, phage therapy, CRISPER-Cas precision therapy

## Abstract

Bacterial drug resistance is rapidly developing as one of the greatest threats to human health. Bacteria will adopt corresponding strategies to crack the inhibitory effect of antibiotics according to the antibacterial mechanism of antibiotics, involving the mutation of drug target, secreting hydrolase, and discharging antibiotics out of cells through an efflux pump, etc. In recent years, bacteria are found to constantly evolve new resistance mechanisms to antibiotics, including target protective protein, changes in cell morphology, and so on, endowing them with multiple defense systems against antibiotics, leading to the emergence of multi-drug resistant (MDR) bacteria and the unavailability of drugs in clinics. Correspondingly, researchers attempt to uncover the mystery of bacterial resistance to develop more convenient and effective antibacterial strategies. Although traditional antibiotics still play a significant role in the treatment of diseases caused by sensitive pathogenic bacteria, they gradually lose efficacy in the MDR bacteria. Therefore, highly effective antibacterial compounds, such as phage therapy and CRISPER-Cas precision therapy, are gaining an increasing amount of attention, and are considered to be the treatments with the moist potential with regard to resistance against MDR in the future. In this review, nine identified drug resistance mechanisms are summarized, which enhance the retention rate of bacteria under the action of antibiotics and promote the distribution of drug-resistant bacteria (DRB) in the population. Afterwards, three kinds of potential antibacterial methods are introduced, in which new antibacterial compounds exhibit broad application prospects with different action mechanisms, the phage therapy has been successfully applied to infectious diseases caused by super bacteria, and the CRISPER-Cas precision therapy as a new technology can edit drug-resistant genes in pathogenic bacteria at the gene level, with high accuracy and flexibility. These antibacterial methods will provide more options for clinical treatment, and will greatly alleviate the current drug-resistant crisis.

## 1. Introduction

Antibiotic resistance is recognized as one of the most serious global threats to human health in the 21st century [1,2,3]. In 2019, researchers found that 1.27 million people died directly from antibiotic resistance according to analyses of cases related to antibiotic resistance in 204 countries and regions around the world, and another 4.95 million people died associated to antibiotic resistance, most of which died from methicillin-resistant *Staphylococcus aureus* (MRSA), reaching over 100,000 [4,5].

Since Penicillium was demonstrated to inhibit the growth of other bacteria in 1928 [6,7], the antibiotic industry has rapidly developed and successfully saved the lives of thousands of wounded patients, and as such, is regarded as one of the greatest discoveries in human history. In the following decades, plenty of different types of antibiotics were identified and successively applied to clinical treatment. However, a few years after the discovery of Penicillin, the phenomenon of bacterial resistance began to appear. In 1972, MRSA was found in England, the United States, and other countries [8]. In 2008, a metallo-β-lactamase gene *blaNDM-1* with the capability to resist the most widely antibacterial hydrocarbon antibiotics was identified for the first time in *Klebsiella pneumoniae* [9]. In 2015, a new drug-resistance gene *mcr-1* was identified in Enterobacteriaceae of pigs in southern China, with the capability to express drug resistance to polymyxins [10]. In 2017, the WHO published its first-ever list of the deadliest superbugs that threaten human health, covering 12 families of dangerous bacteria that have developed resistance to antibiotics, where the “critical” section refers to three bacteria—carbapenem resistant *Acinetobacter baumannii*, carbapenem resistant *Pseudomonas aeruginosa*, and carbapenem resistant and ESBL-producing Enterobacteriaceae (including *Klebsiella*, *E. coli*, *Serratia*, and *Proteus*)—which are all resistant to multiple drugs, and can elicit a range of serious infections [11].

In a natural environment, bacteria are supposed to constantly compete for survival resources, which equips numerous microorganisms with the evolved chemical substances produced in the process of metabolism that can inhibit or kill other microorganisms [12,13], where the Penicillins are metabolites of *Penicillium*, and cephalosporins are metabolites of *Cephalosporium*, etc. Under the pressure of survival, competitors have also evolved corresponding resistance mechanisms to various antibiotics, and antibiotic-secreting strains often have corresponding resistance mechanisms to protect themselves. Studies have demonstrated the resistance of archaea from 30,000 years ago to β-lactam antibiotics (e.g., penicillin) and aminoglycoside antibiotics (e.g., streptomycin) [14,15]. It is obvious that bacterial drug resistance is developed as a self-protection mechanism that bacteria retain in natural selection, and various resistance mechanisms of bacteria have been developed against antibiotics by long-term evolution, enabling bacteria to escape the action of more antibiotics, which aggravates the problem of bacterial drug resistance.

Up to now, antibiotics have been widely utilized for more than 80 years globally, and there exist thousands of available antibiotics, with hundreds of them commonly applied in clinical practice. However, since the 1990s, the identification of antibiotics has gradually ground to a halt, and most novel antibiotics are optimized and upgraded only on the basis of the original antibiotics, without changes in the drug targets and antibacterial mechanism. In recent years, with the development of bioinformatics, synthetic biology, and other biotechnology, new potential antibiotics have been increasingly discovered, such as Teixobactin (2015) [16], Chimeric peptidomimetic (2017) [17], Arylomycin (2018) [18], Corbomycin (2020) [19], Iboxamycin (IBX) (2021) [20] and so on, the most of which have new drug targets in comparison to previous antibiotics, with the nonspecific targets enabling a broad antibacterial spectrum for these compounds. Furthermore, the bacteriostasis achieved through physical or chemical principles in most of them, to a large extent, prevents the bacteria from obtaining drug resistance through mutation.

At present, the small molecule drugs are still adopted as the first choice of clinical antibacterial drugs, which, however, will bring some issues, as for example, they all act as the metabolic pathways shared by microorganisms, rather than drugs killing specific pathogenic bacteria, thus destroying the microbial ecological balance of the organism [21]. Among the novel treatment methods, antimicrobial therapy based on phage therapy and CRISPR-Cas technology has aroused the increasing interest of researchers. As early as 1921, Bruynoghe and Maisin [22,23] firstly applied phage preparations in treating skin infections caused by *Staphylococci.* spp. Since then, phages have been widely adopted in the treatment of otolaryngology, stomatology, ophthalmology, dermatology, and lung diseases. Despite the fact that after the 1940s, phage therapy had faded out due to the gradual popularization of antibiotics, it has returned as a part of the clinician’s weaponry with the increasingly serious problems of antibiotic resistance and the development of new antibiotics falling far behind in recent years, with multiple successful cases of superbugs treated by bacteriophages having been reported in clinical practice [24,25,26]. CRISPER-Cas, as a new gene editing technology, has a high targeting efficiency and simple primer design and multiple other advantages, which are utilized to specifically kill target pathogenic bacteria or knock out drug-resistant genes in the genome. They target and cut the precise sequences in the bacterial genome in a species-specific way to produce the narrowest antimicrobial spectrum possible, so as to achieve targeted sterilization or bacteriostasis. Both methods have developed to be one of the hot issues in the current research on pathogenic bacteria treatment.

## 2. Mechanism of Antibiotics Resistance

Plenty of antibiotic resistant bacteria have continuously entered people’s vision since the discovery of the first antibiotic penicillin [12], triggering a long arms race between humans and bacteria. Despite the multiple natural and synthetic antibiotics that have been added to the battlefield, corresponding strategies will always be identified by bacteria to weaken the lethality of antibiotics. Moreover, additional functions also exist in a large amount of antibiotic resistance mechanisms in the metabolism process of bacteria. For instance, the efflux pump that transports specific antibiotics outside the cell membrane can also pump out toxins such as heavy metal ions to protect cells [27]. Facing the action of antibiotics, the related mechanisms have been continuously evolved by bacteria to resist antibiotics. Additionally, researchers have also discovered new resistance mechanisms to antibiotics in bacteria, involving entering the dormant state, secretion of target-protecting proteins, and regulation of metabolism and initiation of self-repair systems, which together constitute the bacterial defense system against antibiotics. In this chapter, the mechanism of bacterial resistance to antibiotics is mainly summarized (Figure 1).

### 2.1. Target Modification or Mutation

Combination with the target site is required for antibiotics to exert an antibacterial effect, in which the mutation or modification of the target site will intervene with the normal combination, thus affecting the effect of antibiotics (Figure 1a). The frequency of spontaneous mutations in antibiotic resistance is about 10^−8^–10^−9^, which means that one in 10^8^–10^9^ bacteria will develop resistance through mutation [28]. The mutations occur randomly bound to the DNA replication process, and most are detrimental to the host bacteria, which will not be inherited at the cellular or population level. However, when exhibiting evolutionary advantage, the mutations may develop to be dominant through horizontal or vertical transmission. With the huge number of bacteria and the horrendous reproduction speed, the mutation of genes will correspondingly exhibit a high frequency, which allows bacteria to quickly acquire drug resistance through population evolution and horizontal gene transmission (HGT).

At present, it is a consensus that the resistance of *Mycobacterium tuberculosis* (MTB) to rifampicin (RFP) is mainly caused by the mutation of MTB *rpoB* gene [29]. The *rpoB* gene contains an open reading frame of 3534bp and encodes 1178 amino acids (AA), but many experimental studies have confirmed that its mutation mainly occurs in the 81-base region of position 507–533, which is called rifampicin resistance determining region (RRDR) [30,31]. However, some scholars believe that the mutation of RRDR external sequence plays a critical role in the resistance of MTB to RFP [32,33]. Penicillin-Binding Proteins (PBPs) [34] are located on the bacterial cytoplasmic membrane with roles in the synthesis of cell wall peptidoglycan, acting as the target of β-lactam antibiotics. When the mutation occurs, the affinity between β-lactam antibiotics and their target PBPs will disappear, resulting in the failure of the antibiotics to bind to the target, inducing bacterial resistance. A related example of target site modification is the structural alteration of PBPs in MRSA, where the resistance of *S. aureus* to methicillin results from the acquisition of an exogenous gene encoding PBP2a, called *mecA*, which is considered to have low affinity for most β-lactams as a PBPs enzyme. Thus, acquiring *mecA* renders most β-lactam antibiotics ineffective against MRSA. The similar target modification is also showed in bacteria resistance to vancomycin, macrolides, lincosamides, and streptavidin antibiotics [35].

### 2.2. Permeability Reduction

In Gram-negative bacteria (GNB), the cell wall is mainly constituted of proteins and lipopolysaccharides, in which the hydrophilic compounds are hard to pass through the lipid bilayer and must be facilitated by porin channels or outer membrane porins (Omps) [36,37]. Each type of bacteria produces specific porins (e.g., OmpF, OmpC, and OmpE), and the deletion or damage of one or more Omps is one of the sources for bacterial resistance [38]. For example, the loss of OprD porin on the outer membrane of the cell elicits the inefficiency or weakness against *P. aeruginosa* for many broad-spectrum antibacterial drugs, with which the antibacterial drugs cannot enter the cell, leading to the natural resistance to antibiotics (Figure 1b) [39].

After the exposure to antibiotics, the acquired drug resistance can be produced by changing the properties and quantity of porin to reduce the membrane permeability of bacteria. Normally, the channel proteins of bacterial outer membrane constitute non-specific transmembrane channels with OmpF and OmpC, allowing antibiotic and other drug molecules to enter the bacteria [40,41,42]. However, when bacteria are exposed to antibiotics more often, the mutations will be induced in the structural gene encoding OmpF protein, resulting in the reduction in, or loss of, OmpF channel protein, thus preventing the antibiotics such as β-lactams or quinolones to enter the bacteria normally. Gram-positive bacteria have no outer membrane to restrict the entry of drugs, and the outer membrane of mycobacteria is equipped with high lipid content, making the hydrophobic drugs (such as rifampicin and fluoroquinolone) easier to enter cells, while limiting the entry of hydrophilic drugs. Dong et al. [43] found that most strains with a higher resistance to β-lactam are accompanied by mutations of the OmpF-related gene of the membrane channel protein. The inactivation of the structural gene of the OmpF protein can decrease the membrane permeability of bacteria, intervening with the β-lactams, quinolones and other drugs to enter the bacteria, resulting in the acquired drug resistance.

### 2.3. Efflux Pumps

Depending on the antibiotic or toxin challenge, efflux may serve as the most rapid acting and most effective resistance mechanism in the bacterial repertoire of stress responses [44]. The bacterial efflux pump system [45] that has evolved in bacteria is a self-protection mechanism to prevent the accumulation of toxic compounds in cells, which can pump these harmful molecules out of the bacteria (Figure 1c). Bacterial efflux pumps (Eps) located in the plasma membrane of bacteria serve as the transporters to actively expel various substrates from the cytoplasm [46]. Among various families of transporters, several involve the prominent members of efflux transporters: the RND (resistance nodulation and cell division) that are especially crucial in bacteria; MFS (major facilitator superfamily); MATE (multidrug and toxic compound extrusion); SMR (small multidrug resistance); and ABC (ATP-binding cassette) superfamilies or families [47,48]. ABC efflux pumps (recognized as “primary active transporters”) eliminate substrates by consuming the energy generated by ATP hydrolysis, while “secondary active transporters” (MATE, MFS, RND and SMR) utilize proton motive force (PMF) as an energy source by pumping Na and hydrogen out of the membrane [49].

Currently, the efflux pumps identified in Gram-positive bacteria involve members of the MATE family and MFS family, where the MFS family is a characteristic efflux pump [50,51]. Efflux pumps identified in GNBare are widely distributed and may source from all the five families, with the most significant pumps in the clinic belonging to the RND family [44]. The RND efflux family members existing in many GNBand are involved in the efflux of antibiotics, heavy metals, toxins and many other substrates, some of which are specific, for example, Tet pumps tetracyclines or Mef pumps macrolides. Other RND pumps are mostly capable of delivering a wide range of drugs, such as the MexAB-OprM pump in *P. aeruginosa*, which confers intrinsic resistance to β-lactams, chloramphenicol, tetracycline, trimethoprim, sulfamethoxazole, and some fluorine Quinolones [52].

A lot of active efflux systems are nonspecific, which leads to multidrug resistance. For example, the active efflux system (AcorAB-TolC) of *E. coli* can elicit resistance to tetracycline, florfenicol, erythromycin, enrofloxacin and so on. Multiple efflux pump families have been identified in a strain of bacteria, with multiple members involved in each family. For example, the enterobacteriaceae is found to contain the RND efflux family, MFS efflux family and ABC efflux family. The RND efflux family involves AcrAB–TolC and OqxABa –TolC efflux systems, possessing the capability to efflux a variety of antibiotics [53,54]. For more detailed information on efflux pumps, such as the structure and function, it is recommended to refer to the review of DU’s Multidrug efflux pumps: structure, function and regulation [44].

### 2.4. Hydrolase or Inactivating Enzyme

The inactivating enzymes produced by bacteria, such as antibiotic hydrolases or inactivating enzymes, can hydrolyze or modify antibiotics entering the cell to render them inactive before reaching the target site (Figure 1d). There exist plenty of aminoglycoside-modifying enzymes in bacteria, such as N-acetyltransferase, O-phosphotransferase and O-adenosyltransferase, which, respectively, acetylate, phosphorylate or adenylate aminoglycoside antibiotics to transform the structure of antibiotics. Inactivating enzymes produced by bacteria mainly involve: β-lactamase, aminoglycoside inactivating enzymes, chloramphenicol acetyltransferase, etc. [55]. The β-lactamase can covalently bind to the carbonyl moiety of the antibiotic to disrupt its cyclic structure, inducing degradation in the β-lactam antibiotic before reaching the target. It can also rapidly and firmly bind to β-lactam antibiotics through non-hydrolysis, preventing the antibiotics from exerting drug resistance through binding to the target site. β-lactamases are secreted by many bacteria for up to eight different types, each capable of hydrolyzing specific β-lactam rings [56]. Carbapenem and extended-spectrum β-lactamases (ESBLs) are the two most primary β-lactamases. The main mechanism of Enterobacter resistance to carbapenem antibiotics is the production of enzymes that hydrolyze carbapenem. Those Enterobacteriaceae bacteria that can produce carbapenem enzymes are called carbapenem-producing Enterobacteriaceae (CPE). ESBLs can destroy most β-lactam antibiotics, such as penicillin and cephalosporins, but fail to destroy carbapenem antibiotics, and are commonly produced by *E. coli*, *K. pneumoniae*, *P. aeruginosa*, *A. baumannii* and other bacteria [57,58].

Rifampicin is adopted as the first choice for the treatment of tuberculosis and leprosy, of which the antibacterial activity depends on the inhibition of bacterial RNA polymerase. Researchers have identified a group of NAD-dependent enzymes in bacteria, which inactivate rifampicin [59] by transferring an ADP-ribosyl molecule to the hydroxyl group of the long aliphatic carbon chain of the rifampicin structure. Chloramphenicol acetyltransferases (CATs) promoted the acetyl group in acetyl-CoA to covalently link to two hydroxyl groups of chloramphenicol and prevent chloramphenicol from binding to ribosomes, thus exhibiting resistance to chloramphenicol [55]. Aminoglycosides play an antibacterial role by binding to 23SrRNA of bacterial 50S subunit to block protein synthesis. However, several resistance genes have been identified in *S. aureus, Enterococcus faecium, M. tuberculosis*, *E. coli*, *Salmonella.* spp. and other bacteria, such as *lnu (A)* to *lnu (F)* and *linAN2*, which inactivate lincomycin with the encoded nucleotide transferase [60].

### 2.5. Metabolic Alteration or Auxotrophy

Although metabolism has been demonstrated to actively contribute to antibiotic lethality, antibiotic resistance mutations are merely identified in metabolic genes, and metabolic dysregulation does not serve as a commonly cited mechanism of antibiotic resistance. In 2021, James’s team found for the first time that mutations in core genes in some metabolic pathways can induce antibiotic resistance, which are widely present in the genome of clinically pathogenic *E. coli* [61], including the core genes of metabolic pathways, such as the *sucA* gene (2-oxoglutarate dehydrogenase enzyme) involved in catalyzing the tricarboxylic acid cycle. The gene with this mutation reduces basal respiration by inhibiting the activity of the tricarboxylic acid cycle elicited by antibiotics, avoiding the occurrence of metabolic toxicity, inhibiting the killing effect of antibiotics, and eventually leading to antibiotic resistance [62].

The essential metabolic pathways required to synthesize amino acids, nucleotides, vitamins, fatty acids or metabolic coenzymes at the genetic level are found to be lacking in auxotrophs [63,64,65]. Microbial communities are composed of cells with varying metabolic capacity, regularly including auxotrophs lacking essential metabolic pathways. In contrast to prototrophs that can flexibly switch between metabolite synthesis and uptake, the growth of auxotrophs is constitutively dependent on the extracellular availability of these metabolites [66,67]. Sulfonamides possess a similar structure to that of p-aminobenzoic acid (PABA), which inhibits the activity of dihydrofolate synthase and prevents folate metabolism by competing with PABA to bind to the active site of dihydrofolate synthase in the process of bacterial folate metabolism. As folic acid is the precursor of nucleic acid synthesis, its deficiency will hinder nucleic acid synthesis and inhibit bacterial growth and reproduction [68]. However, bacteria can weaken the inhibitory effect of sulfonamide antibiotics on folic acid metabolism through metabolism enhancement, and can also obtain folic acid from extracellular in an auxotrophic way to maintain normal metabolism (Figure 1e). In 2022, Markus Ralser’s [69] team have revealed a metabolically imprinted mechanism that links the presence of auxotrophs to an enhancement in metabolic interactions and gains in antimicrobial drug tolerance. Moreover, the elevated efflux activities reduce the intracellular drug concentrations, allowing cells to grow in the presence of drug levels above minimal inhibitory concentrations. These results indicate that auxotrophy is beneficial to alleviate the sensitivity of bacteria to antibiotics, thus reducing the antibacterial effect of antibiotics.

### 2.6. Target Protective Proteins (TPPs)

Bacterial synthetic protein protects some antibiotic targets from a combination of antibiotics, eliminating their bacteriostatic effects (Figure 1f), and Daniel N. Wilson’s team [70] divided target protection into three types according to the mode of action. In Type I target protection, the binding of tetracycline ribosomal protection proteins (TRPPs) to ribosomes can reverse the distorted ribosomal structure, evoking changes in ribosome configuration, and directly interfering with the interaction of tetracycline D-ring and 16S rRNA base C1054. Tetracycline class drugs cannot bind to it and dissociate from the 30S subunit of the binding site, thereby protecting the ribosome, in which 13 TRPPs classes have been identified [61,62]. In Type II target protection, antibiotics are indirectly removed by changes in target conformation. Mediated by antibiotic-resistant ABC-F proteins, this group of proteins is the primary source of clinical resistance to antimicrobials of ribosome 50S subunits, including lincomycins, macrolides, azadones, phenols, pleuromutilins, and stroopogramins of groups A and B [71,72,73]. Type III target protection proteins induce changes in target conformation so that antibiotic targets can also work in the state of binding to antibiotics. In recent years, clinically isolated *S**.aureus* and other staphylococcus resistance to fusidic acid has increased significantly, mainly due to the level acquisition of genes encoding the FusB-type protein. The resistance of FusB resistance proteins to fusidic acid is due to the fact that fusB proteins bind to elongation factor G (EF-G) and drive its dissociation from ribosomes (even in the presence of fusidic acid). Once the elongation factor leaves the ribosome, fusidic acid may be separated from EF-G due to its low affinity for free EF-G [74,75,76].

### 2.7. Initiation of Self-Repair Systems

The multiple antibiotic resistance operon of enteric bacteria manipulates the DNA repair and outer membrane integrity (Figure 1g), which contributes to enhancing the antibiotic resistance. The *E. coli* multiple antibiotic resistance (mar) locus was recognized as a determinant for cross-resistance to tetracyclines, quinolones and β-lactams [77]. Studies have shown that the active efflux mechanism controlled by the global operon is one of the primary reasons for the multiple antibiotic resistance of bacteria. Among them, the multiple antibiotic resistance protein family (Mar family), as a transcriptional regulatory protein, plays an important role in the production of drug resistance, the synthesis of toxic factors and other physiological processes. As the prototype of a multiple antibiotic resistance protein family, *E. coli* MarR protein has a negative regulatory function on MarRAB operon and inhibits the expression of downstream related drug resistance genes [78]. Transcription factors MarR and MarA confer multidrug resistance in enteric bacteria by modulating the efflux pump and porin expression [79,80,81]. In 2017, Sharma’s team demonstrated that MarA upregulates genes required for lipid trafficking and DNA repair, thus reducing DNA damage induced by antibiotic entry and quinolone [82]. The initiation of self-repair systems reduces the rate of antibiotics entering cells and the impact on cell structure and metabolism through gene regulation of the expression of related genes. This method cannot completely eliminate the bacteriostatic effect of antibiotics, but can make bacteria enhance their tolerance to antibiotics.

### 2.8. Changes of Cell Morphology

The mechanism of antibiotics is adapted through mechanical feedback between cell growth and morphology, altering uptake efficiency by modulating relative body area (Figure 1h). The increase in cell volume contributes to diluting the antibiotics entering the bacteria, while both bending and widening can reduce the surface volume ratio so that fewer antibiotics pass through its surface. In 2021, Aaron’s team [83] found that cells of the commonly used model organism *C. crescentus* could regain the growth rates they had prior to stimulation by antibiotics, accompanied with significant morphological transformations. Once the antibiotic was removed, the cells returned to their original shape after a few generations. That bacteria change shape to avoid being targeted by antibiotics was also previously demonstrated by another team [84], in which, however, the bacteria sloughed off their entire cell wall to avoid the drug, resulting in a shape distortion. In Aaron’s study, the cell walls remained intact, but stretched so violently that a “C” shape was formed. Bacteria are able to decrease the time it takes for antibiotics to exert biological effects in this physical way and increase the concentration of antibiotic tolerance. Using single-cell experiments and theoretical models, they proved that the change of cell morphology is a feedback strategy to enable it to adapt to the antibiotic environment and survive. The bacteria after “Metamorphosis” can overcome the pressure of antibiotics and recover to the state of rapid growth [83].

### 2.9. Biofilm Protection

We have summarized eight different drug resistance mechanisms of bacteria at the individual level. However, in the actual environment, the vast majority of bacteria coexist in the form of communities, jointly resisting the effects of antibiotics in a collective form, with the biofilm serving as a critical form of protection (Figure 1h). Bacterial biofilm [85] is a special survival form established by bacteria adsorbed to inert objects such as medical materials or the surface of the body’s mucosa, in which the protein is surrounded by an autocrine polymer matrix. Dense biofilms are constituted to provide exposure protection for their members, forming physical barriers to limit the diffusion of antibiotics into the population and enhance the protection provided by antibiotic inactivation. In addition, due to the gradient of nutrients and oxygen, the decrease in the metabolic activity of the biofilm’s center enables the biofilm to induce a tolerant cell state, thus elevating the proportion of persistent cells in the population. Biofilms can also enhance the drug resistance by altering the expression of pre-existing ARG [86]. In contrast to single-species biofilms, the inter-species interactions among multi-species biofilms can further enhance the collective by altering the spatial structure of biofilms, promoting the expression of resistance mechanisms and allowing individually expressed antimicrobial defenses to protect entire communities. The resistance mechanism formed in the way of antibiotic resistance, collective tolerance or exposure protection to antibiotics is not specific and serves as the first line of defense for bacteria to develop resistance to antibiotics.

Bacterial communities can survive antibiotic exposure through interspecific interaction: (1) collective drug resistance, that is, the interaction within the community can enhance the capability of its members to resist antibiotics to continue to grow, thus elevating the MIC of the community; (2) Collective tolerance, i.e., interactions within the community can alter cellular states, such as retarding metabolism, so as to temporarily reduce the rate of cell death during antibiotic treatment without increasing the MIC; (3) Contact protection to protect the interaction of its sensitive members by reducing the effective concentration of antibiotics in the community [87,88]. In the mixed biofilm, *P. aeruginosa* can elicit the metabolic transformation of *S. aureus*, inhibit its growth and provide *S. aureus* with protection against vancomycin [89]. Correspondingly, *S. aureus* can enhance the tolerance of *P. aeruginosa* to tobramycin by promoting aggregation and altering the biofilm structure in the CF model system [90]; The interspecific signal transduction of indole secreted by *E. coli* activates the expression of indole dependent multidrug efflux pump in *P. putida*, while Pseudomonas itself cannot produce indole, which results in an elevation in the resistance level of *P. putida* [91]. Similarly, *S. maltophilia* is a Gram-negative bacterium, generally appearing accompanied by *P. aeruginosa* during bacterial lung infection. It can diffuse the secretion of signal factors, transform the biofilm structure of *P. aeruginosa*, and stimulate the synthesis of proteins so as to provide resistance to cationic antimicrobial peptides, such as polymyxin [92].

## 3. Antibacterial Methods

Drug resistance of bacteria has become a major challenge in the global public health field, especially the nosocomial infection caused by some MDR bacteria, which brings more difficulties to clinical treatment. Faced with this challenge, scientists began to try to develop new antibacterial methods, some of which showed strong antibacterial effects in the experimental research stage and showed great clinical application potential, such as new antibacterial compounds molecule, phage targeted elimination of MDR bacteria, CRISPR-Cas system targeted elimination of MDR bacteria, etc. These methods have their own unique advantages, which can meet the diverse treatment requirements of different pathogenic bacteria. In this section, we mainly reviewed the research progress and antibacterial mechanism of the above three antibacterial methods.

### 3.1. Newly Potential Bacteriostatic Compound Molecule

Traditional antibiotics show rather limited power facing the super-resistant bacteria such as ESKAPE (*E. faecium*, *S. aureus*, *K. pneumoniae*, *A. baumannii*, *P. aeruginosa*, and *Enterobacter* spp.), therefore the research and development of new antibiotics is imminent. Antibiotics mainly target the essential functions of bacteria, covering the synthesis of nucleic acid and protein and metabolic pathways, etc. The drug resistance can be acquired through gene mutation of the target. However, with the efforts of many scientists, more and more new antibiotics and antibacterial mechanisms have been discovered, which seem to be making it difficult for bacteria to obtain drug resistance through mutation (Table 1).

Teixobactin was discovered by Professor Kim Lewis and his colleagues in 2015 [93], which is considered the first novel antibiotic discovered in the last 30 years. It can kill a lot of deadly pathogens, such as MRSA and Vancomycin-resistant *Enterococcus* (VRE), and also treat many common infections, such as tuberculosis and septicemia. More significantly, unlike most other antibiotics that mainly attack bacterial protein, it kills bacteria by destroying their cell walls, making it difficult for pathogens to develop drug resistance to it. Studies have found that the Teixobactin compound Leu10-teixobactin can destroy membrane lipids coated with bacteria, involving glycerophosphates and fatty acids. It also intervenes in the metabolism of peptidoglycan (lipids I and II) and the biosynthesis of cell wall teichoic acid (lipid III). In 2017, Maffioli’s team reported a new antibiotic Pseudouridimycin [94], acting on nucleic acid metabolism, which is the first nucleoside analogue inhibitor that has been discovered. It can destruct the binding of nucleoside triphosphate (NTP) to RNA polymerase by occupying the binding site of NTP. Binding to different sites to inhibit bacterial growth from rifampicin, this antibiotic does not promote cross-resistance of bacteria to rifampicin.

In 2018, Christopher Heise’s team [18] discovered a synthetic derivative of arylomycin called G0775, which serves as a macrocyclic lipopeptide to inhibit type I signal peptidase (SPase). SPase is an essential bacterial membrane-bound protease that clears the N-terminal signal peptide of ectopic proteins during secretion. Without the cleavage of this signal peptide, these key proteins cannot be delivered to cells outside, but accumulates in the membrane and kills the bacteria [103,104]. In the same year, Christopher and his team screened out a small molecule inhibitor named G907 from three million drug candidates. The inhibitor with a small molecule can specifically inhibit *E. coli* MsbA (MsbA is an essential ABC (ATP-binding cassette) transporter involved in lipid A transport across the cytoplasmic membrane of GNB), thus the MsbA fails to complete the key conformational transition required for physiological functions. Inconsistent to conventional ABC transporter inhibitors, G907 and its analogues will not competitively inhibit the transport substrates, but affect the function of MsbA through structural inhibition [95]. 

In 2019, Lewis’ team [96] discovered a new antibiotic, called darobactin, a metabolite of Photorhabdus with a short peptide of seven amino acids that selectively kill GNBby binding to a key outer membrane protein, BamA. A novel bactericidal mechanism is proposed in the article, in which the bacterial outer membrane is disrupted and the cell lytic death is induced with the BamA outer membrane protein inhibited. In the same year, Professor John A. Robinson’s team and their partners [17] reported a series of chimeric peptidomimetic antibiotics with broad-spectrum antibacterial activity against GNB, with the new mechanism of their “shell attack” explained. Chimeric peptidomimetic covers a class of synthetic antibiotics inspired by scaffolds derived from natural products, containing a β-hairpin peptide macrocycle linked to the macrocycle found in the polymyxin and colistin family of natural products. They are bactericidal with an action mechanism involving binding to both lipopolysaccharide and the main component (BamA) of the β-barrel folding complex (BAM) that is required for the folding and insertion of β-barrel proteins into the outer membrane of GNB.

In 2020, a new glycopeptide antibiotic Corbomycin was discovered by Wright and his team [19], which played a bactericidal role by blocking the effect of cell autolysin on cell wall and preventing the collapse of the cell wall. Unlike the β -lactam antibiotics and glycopeptide antibiotics, which prevent the synthesis of a cell wall by seizing the binding sites, this new antibiotic forms thicker and denser cell walls around the bacteria to inhibit bacterial division and prevent the normal reproduction of bacteria. In the same year, MIT researchers [97] identified a powerful new antibiotic compound using a machine-learning algorithm, which killed a large amount of the world’s most problematic disease-causing bacteria in laboratory tests, covering some strains that are resistant to all known antibiotics. Preliminary studies suggest that Halicin kills bacteria by disrupting their capability to maintain an electrochemical gradient across the cell membranes. Among other functions, this gradient is required to produce ATP (molecules that cells use to store energy), which means the cells will die if the gradient breaks down. This type of killing mechanism could be obstructive for bacteria developing resistance. Zemer Gitai and his colleagues [98] discovered an antibiotic SCH-79797 with dual mechanisms, which attacks bacteria through two different mechanisms within a molecule, like a “poisonous arrow” piercing the cell wall of bacteria and destroying folic acid in the cell. To prove the prevention of drug resistance of SCH-79797, Martin et al. carried out countless different assays and methods, including a “continuous passage” experiment over 25 days, in which the bacteria still failed to develop drug resistance. In 2021, Nudler’s team [100] found that the inhibitory effect of antibiotics can be effectively enhanced by attacking the pathogen defense system—the H_2_S biogenesis system. Studies have found that cystine sulfate lyase (CSE) is the main source of H_2_S production by *Staphylococcus aureus* and *Pseudomonas aeruginosa*, the H_2_S product will be absent when the pathogen lacks bCSE (bacterial CSE). Therefore, the inactivation of CSE will stimulate the sensitivity to bactericidal antibiotics of these two bacteria, which will be killed by antibiotics. Taking bCSE as the target, the researchers screened out three bCSE inhibitors from 3.2 million small molecule compounds, named NL1, NL2 and NL3, respectively, which were experimentally proved to inhibit the production of bacterial H_2_S to enhance the bactericidal efficacy of antibiotics. Andrew’s team [20] reported a new synthetic antibiotic iboxamycin (IBX) that can shift methylated ribosomal nucleotides and expose drug binding sites, so that drugs can still bind to ribosomes and perform efficacy when ribosomes have been methylated. Brady’s team [99] developed a “tracing to the source” synthesis method, in which the corresponding genes were directly searched in the bacterial genome, the structure of its natural products was predicted, and finally the chemical synthesis was performed. A total of 35 groups of biosynthetic gene clusters (BGC) were found to possibly encode polymyxin antibiotics, of which the Macolacin expressed perfect inhibitory activity against Gram-negative pathogens carrying the *mcr-1* resistance gene. The bioinformatics technology used to predict and analyze big data contributed to quickly identifying new antibiotics. In the same year, Brady’s team [101] found another new antibiotic that can inhibit MRSA with the similar method, called menaquinone-binding antibiotic (MBA), which targets menaquinones that plays a key role in the electronic transmission of bacteria.

In 2022, Brady’s team [102] synthesized a new cyclic non-ribosomal lipopeptide antibiotic cilagicin depending on the computer model of bacterial gene products. It works by combining undecaprenyl phosphate (C55-P) and undecaprenyl pyrophosphate (C55-PP) that contribute to maintaining bacterial cell walls. Existing antibiotics like bacitracin will bind to one of these two molecules, but never at the same time, and bacteria can often resist this drug by piecing together a cell wall with the remaining molecules. Brady’s team speculated that cilagicin’s capability to simultaneously bind these two molecules might become an insurmountable obstacle with regard to preventing drug resistance.

### 3.2. Phage Targeted Elimination of MDR Bacteria

Bacteriophages are viruses with the capacity to infect bacteria, fungi, algae, actinomycetes or spirochete and other microorganisms, which can be widely found in the natural environment, soil, ocean and sewage, with a large number and variety, ranked as the most abundant and common organisms on earth [105]. The number of phages in nature reaches 10^31^, which is 10 times the number of bacteria. A study revealed more than 54,000 bacteriophages in the human gut, 90% of which are unknown to scientists [106]. Phage therapy refers to phages being made into microbial agents to treat pathogenic bacterial infections, and they only prey on certain bacteria in the human body due to the strong specificity of bacteriophages. With a reduction in bacteria, bacteriophages are recognized as exogenous substances and swallowed and decomposed by the immune system of the body. Phages must parasitize in living bacteria with strict host specificity, which is determined by the molecular structure and complementarity of phage adsorbed organs and receptors on the surface of receptor bacteria. It is unique in that it replicates and proliferates in the host bacteria through parasitism, producing numerous offspring phages, and finally “eating” the bacteria. New phages spread around to searching for new hosts and adsorption, repeating the above cycle, and finally eliminating the target infected bacteria (Figure 2). In 2022, Jessica Little’s research team [107] proved that the bacteriophage therapy, combined with antibiotics and surgery, can effectively treat the infection of *M. chelonae* in patients with a low immune function. This is the first to report the successful treatment of *M. chelonae* infection by bacteriophage therapy, with the observed clinical efficacy described.

#### 3.2.1. The Function of Phage Endolysin

The endolysin of phage is a lyase produced in the later stage of lysis after infecting bacteria [110,111], with the main function being to assist the release of new bacteriophages from infected cells by degrading bacterial peptidoglycan [112]. Phage endolysins are analogous to bacterial lysins in structure and function, with close association to the small family of mammalian peptidoglycan recognition proteins [113]. Endolysins are considered a promising antibiotic with numerous unique advantages over common antibiotics, including strong specificity for the host, lower risk of bacterial resistance, resistance to biofilms and biological effect on biofilm and capsule surface, etc., [114,115,116]. Endolysins spread in various types, most of which are species-specific, and a large number of studies have demonstrated their critical role in antimicrobial and anti-biofilms. According to the mode of action, EADs are categorized into three groups: (a) glycosidases, cleaving the glycan portion of peptidoglycan (MurNAc-GlcNAc); (b) amidases, cleaving the amide bond between the glycan moiety (MurNAc) and the peptide moiety (L-alanine); (c) endopeptidase, cleaving the peptide bond between two amino acids of the stem peptide [117,118]. The enzymatic activity of endolysins is influenced by the composition of cell walls. As for Gram-positive bacteria, endolysins readily access the peptidoglycan layer and hydrolyze the basic bonds of peptidoglycan, resulting in osmotic lysis and cell death [119]. However, the cell wall of GNB is equipped with an outer membrane with a lipopolysaccharide layer, which avoids giving the endolysins access to the peptidoglycan layer [120].

It has been mentioned that biofilm plays a significant role in enhancing bacterial resistance to antibiotics and the evolution of drug resistance. Numerous studies have proved the good effect of Endolysin on destroying the structure of biofilm. For example, the staphylococcal endolysins SAP-2 and Phi11 eliminate biofilms on polystyrene surfaces [121,122], while the endolysin LysH5 has staphylococcal biofilm-removal properties, without resistant cells after treatment [123]. PlyGRCS, a staphylococcal endolysin destroying MRSA, disturbs biofilms as well [124]. The endolysin Lys84 with two catalytic domains (CHAP and amidase_2) and a CBD (SH3b) effectively removes around 90% of the biofilms of *S. aureus*, with CHAP and Amidase_2 domains remaining 61.20% and 59.46% of lytic activity as well as 84.31% and 70.11% of the anti-biofilm activity of Lys84, respectively [125]. Other research revealed that the amidase domain of the *Listeria monocytogenes* phage vB_LmoS_293 endolysin prevented biofilm formation on abiotic surfaces [126], while the Salmonella endolysin Lys68 could decrease biofilms in combination with malic or citric acid [127]. For more detailed information on the bacteriostasis and biofilm inhibition of Endolysin, please refer to Wang’s review [128]. By being linked to other domains, endolysins can target intracellular bacteria as they are delivered across the Gram-negative outer membrane or into eukaryotic cells [129]. These findings indicate that endolysins can serve as promising anti-biofilm agents. In fact, endolysins are more suitable for therapy in comparison to bacteriophage with the advantages in safety, quality control and policy. In addition, these endolysins may be further modified to elevate their specificity or effectiveness in eradicating different microorganisms.

#### 3.2.2. Phage and Antibiotic Combination Therapy

When phages and antibiotics are utilized in combination, phages with drug efflux pumps as receptors force the efflux pumps of DRB to mutate to develop phage resistance, resulting in an increase in susceptibility to antibiotics [130,131]. Benjamin K Chan’s team [130] found that phages can take the outer membrane protein OprM of the MexAB and MexXY multidrug efflux systems as receptor binding sites. In order to prevent phage invasion, *P. aeruginosa* resistant to multiple drugs varies its efflux pump protein structure, increasing its susceptibility to several antibiotics. The TolC protein of *E. coli* is part of the bacterial efflux system, as well as a receptor for phage entry into cells, and TolC-altered resistant mutants are resistant to phage but highly sensitive to neomycin [132]. These phenomena are all related to the drug efflux pump of phage-mutant-resistant bacteria.

The bacterial capsule is rather critical for the adsorption of phage, of which the deficiency makes the bacteria easily change the phage adsorption efficiency, leading to the re-sensitization of antibiotics to DRB [25,133,134]. Schooley’s team [24] found that multidrug-resistant *A. baumannii* can obtain resistance to phages by losing its capsule, but this makes it easier for antibiotics to penetrate its outer membrane. Altamirano’s team [135] demonstrated that the genetic deletion of phage-resistant *A. baumannii* synthetic capsules disrupted phage adsorption but became re-susceptible to antibiotics. The formation of biofilms contributes bacteria to evading the host’s immune system and increases their antibiotic resistance, with close association with severe infections by a variety of bacteria [136]. Mature biofilms can prevent antibiotics from penetrating into the membrane, while phages can form channels in the biofilm, allowing antibiotics to diffuse into the biofilm and reach a higher concentration [133]. In order to penetrate the exopolysaccharides (EPS) layer more effectively, some phages will carry an EPS polysaccharide depolymerizing enzyme on the tail or tail fiber [134], which can degrade biofilms showing synergistic effects with antibiotics, and some can also kill bacteria in biofilms [137,138,139].

### 3.3. CRISPR-Cas System Targeted Elimination of MDR Bacteria

Traditional antibiotics will leave sublethal doses in the process of clinic treatment, and bacteria can resist the effects of antibiotics through phenotypic changes mediated by preexisting genetic spectrum. It takes a certain period for microorganisms to obtain drug-resistant genotypes, which results in the rapid appearance of drug-resistant pathogens [140]. Therefore, in order to meet the requirements of current antibacterial treatment, this new antibacterial method must be used to help further shape the following two standards: (1) narrow-spectrum antibiotics, that is, targeting specific pathogens; (2) the capability to rapidly enhance the response with the emergence of new DRB [141]. The CRISPR-Cas system involves CRISPR array and a group of genes encoding Cas protein [142], among which CRISPR array is composed of a leader sequence, repeat sequence and spacer sequence, with mature crRNA formed by its transcript after the related protein is processed. The nucleic acid-protein complex formed by being combined with Cas protein recognizes and degrades DNA or RNA [142]. Researchers found that the modification of the guide RNA (GRNA) sequence of the CRISPR-Cas system can target the paired DNA or RNA sequence. This technology has been widely adopted in gene editing and regulation, showing unique advantages and strong potential in fighting superbugs. Currently, the principle of this method is basically consistent in cells, most of which are an exogenous introduction of designed sgRNA and Cas sequences. Without repair templates in bacteria, DSB caused by CRISPR-Cas cannot be repaired, which will induce the death of bacteria or the loss of target fragments [143,144,145], with the difference lying in the different delivery vectors. At present, the following three vectors are widely taken in experimental research, namely plasmid vector (conjugated transfer, non-conjugative transfer), phage vector and nanoparticle vector (Figure 3).

#### 3.3.1. Plasmid Vector

According to the characteristics of horizontal transfer, plasmids can be divided into non-conjugated plasmids and conjugated plasmids. The former can enter the host bacteria by natural transformation without mediation of donor bacteria. While the latter enters the recipient bacteria from the donor bacteria through its own type IV secretion system under natural conditions [146]. People make the CRISPR-Cas system target and cut drug-resistant genes by integrating the sequences encoding the CRISPR-Cas system or some of its components on plasmid vectors, thus reducing the resistance of DRB to antibiotics. Bikard’s team [143] transferred the CRISPR01 sequence of *Streptococcus pyogenes* SF370 into *S. pneumoniae* R6 using non-conjugation plasmid pCEP, which contains DNA that can transcribe tracrRNA, and four cas genes referring to cas9, and several repeat-spacer units. The crRNA generated by the spacer sequence can target the drug-resistant gene on *S. pneumoniae* chromosome and kill the host bacteria carrying the drug-resistant gene. Transfer efficiency is the main factor limiting the clinical application of the CRISPR-Cas system, and elevating the plasmid conjugation efficiency can expand the application prospect of this method. *P. aeruginosa*, as a serious pathogenic bacterium inducing lower respiratory tract infection in clinic, exhibits a high drug resistance level. Plasmids involving spacer sequences and editing templates of targeted resistance genes are introduced into the bacteria, and the crRNA formed by its transcription guides the endogenous I-F CRISPR-Cas system of *P. aeruginosa* to remove the resistance genes or plasmids, thus reducing the antibiotic resistance level of the bacteria [147].

#### 3.3.2. Phage Vector

Compared with the plasmid vector, the phage vector not only has a strong capability to infect host bacteria, but can also carry larger DNA fragments, which can be introduced into the CRISPR-Cas system encoding multiple proteins. In addition, the nucleic acid coated by phage protein is stable and difficult to be degraded. Therefore, the phage vector can also be adopted to prevent and control drug resistance gene transfer by the CRISPR-Cas system. Kim’s team [148] introduced sgRNA and Cas9 based on the conserved sequence of β-lactamase mutants into extended-spectrum β-lactamase-resistant *E. coli*, and successfully realized the inactivation of more than 200 β-lactamase gene mutants in pathogenic bacteria using phage as vector, promoting the DRB to recover their sensitivity to β -lactamases antibiotics. Yosef’s team [149] adopted lysophage to wrap the type I CRISPR-Cas3 system containing six cas genes and a plurality of resistance gene spacers targeting to *ndm-1* and *ctx-M-15*. This lysophage can selectively destroy drug-resistant plasmids and restore the host bacteria’s sensitivity to multiple antibiotics. Lu Guanda [150] designed several RNA guiding chains that target drug-resistant genes in bacterial genomes, involving the gene encoding *ndm-1* enzyme. When the CRISPR system was utilized to fight against *ndm-1*, the results showed that this method specifically killed more than 99% of bacteria carrying *ndm-1*, and successfully targeted another gene encoding *bla_SHV-18_*.

#### 3.3.3. Nanoparticle Vector

Nanoparticle generally refers to nano-scale carriers produced from of high-molecular polymer or inorganic materials, which are small in size with strong biofilm penetration capability. Antibacterial drugs can be encapsulated in polymer carriers in nanoparticles, released in vivo to play a bactericidal role with high solubility [151]. DNA coated or adsorbed by inorganic nanoparticles (such as liposomes and cationic polymers, etc.) can overcome the barrier outside the cell by entering the cell through endocytosis to release, providing a new idea for the delivery of therapeutic drugs into the body. Kang’s team [152] covalently modified the Cas9 protein of *S. pyogenes* with branched Pei (bPEI), and assembled it with sgRNA targeting to *mecA* into a nano-sized complex, thus realizing the targeted cutting of drug-resistant genes in the genome of MRSA. Cas9 protein modified by bPEI can maintain its nuclease activity for a long time and further enhance the cleavage efficiency of double-stranded DNA.

CRISPR-Cas as a method to fight bacteria at the gene level has numerous unique advantages, e.g., realizing precise targeting and directional transformation of pathogenic bacteria in microbial communities: (a) in the first strategy, CRISPR-Cas technology targets specific strains (covering drug-resistant and sensitive strains) to inactivate them; (b) targeting drug-resistant strains to inactivate cells, targeting genomes to inactivate cells; (c) targeting drug-resistant genes to restore their sensitivity, and random genes. The method significantly enriches the selectivity of clinical treatment schemes, performing directional transformation on microbial communities [153]. However, CRISPR-Cas antibacterial agent is still in its infancy, facing various challenges in drug design and delivery, e.g., while in bioplasm electrolysis is the method of choice to introduce the CRISPR-Cas system into the bacterial cells in the vast majority of the studies, it would not always be possible to perform in vivo. In different delivery mechanisms the introduction of biological macromolecules is required (nucleic acid or the complex of protein and nucleic acid) into bacteria, which is both difficult and easily degraded by intracellular protease or nuclease. There are a lot of unknown obstacles in every link from laboratory in vitro test to animal test, and ultimately to human test. Therefore, it still requires a lot of work to ensure that the vehicle reaches the infected site and effectively delivers the CRISPR-Cas load to the pathogen at the infected site.

## 4. Discussion

Due to the extensive utilization of antibiotics, the environment that bacteria live in has varied dramatically, in which the DRB adapting to the new environment selectively survived. This “natural” selection will push the bacterial population to evolve in the direction of enhancing drug resistance. In this review, we summarized different mechanisms of drug resistance, of which the source can be divided into “mutation acquisition” and “contact acquisition”. The former is formed with gene mutation during the long-term evolution of bacteria, without anything to do with whether bacteria having been exposed to antibiotics. The selective effect of antibiotics expands the frequency of drug-resistant genes in the population. The latter is the horizontal transfer of drug resistance genes to promote bacteria to acquire drug resistance, which can occur between the same bacteria or different bacteria [154,155], acting as a critical route for bacteria to obtain drug resistance genes. Regardless of the source, most of the drug resistance genes originate from gene mutations, with the capability to be inherited. In addition, different types of bacterial drug resistance mechanisms endow bacteria with different drug resistance. For example, due to the difference in cell structure between GNB and Gram-positive bacteria, the outer membrane of GNB is more likely to acquire natural resistance to antibiotics through the mutation of the outer membrane pore protein, but also as a result of the presence of lipopolysaccharide (LPS) in the outer membrane of GNB, it becomes the target of colistin. Some drug resistance mechanisms exist specifically in bacteria, for example, changing cell morphology to resist the action of antibiotics. At present, there are only relevant reports in *C. crescentus*, but there may be no corresponding drug resistance mechanisms in other bacteria.

With the development and deep utilization of AI (artificial intelligence), deep learning, big data analysis and other technologies, potential new antibiotics can be mined with more approaches, greatly accelerating the development of new antibiotics. For example, Halicin [97], a powerful new antibiotic molecule, was identified from more than 107 million molecules. This brand-new action mechanism of inhibiting bacteria by affecting transmembrane potential, and the strongest molecule selected by AI has no link with the traditional drug-target interaction, providing us more inspiration when developing new antibiotics. These new technologies break the traditional limitations in drug research and development, which are expected to develop new antibiotics from new perspectives. In addition, in the past fifty years, the identification and development of antibiotics have been reliant on the semi-synthetic chemical modification of natural products, which currently fails to conform to the rapidly evolving threat of bacterial drug resistance, while the total synthetic chemical modification with reasonable design can easily solve this difficulty. Teixobactin [16] and Iboxamycin [20], two potential antibiotics, have been fully synthesized, which is of great significance for exploring the molecular mechanism of their antibacterial activity and further developing new compounds. The recently reported new antibiotic Cilagicin [102] was obtained by computer model prediction and biosynthesis, which provided an efficient method for the discovery of new antibiotics in the future.

A great proportion of antibiotics with good bacteriostatic effect in the experimental stage may eventually fail to enter clinical treatment due to their failure to pass a series of pre-clinical trials or clinical trials. As a new type of treatment, phage therapy has accumulated a large number of successful cases so far, which possess a lot of incomparable advantages in comparison to the traditional antibiotic treatment, however, multiple aspects are still to be modified when it comes to technology, methods and policies. It is worth believing that these problems will be solved in the future and phage therapy will play a critical role in the fight against “superbugs”. At present, CRISPER-Cas gene editing technology still remains in the experimental research stage, without cases that have been successfully applied to clinical treatment. Despite many problems in technology and application, this technology may completely rid mankind of the threat of pathogenic bacteria with the deepening of research. In addition, there are various other new antibacterial methods with the hope of defeating DRB, such as nanomaterials for antimicrobial-free antimicrobial applications, antimicrobial nano-pharmaceuticals [156,157], light-mediated antimicrobial nanomaterials [157,158], catalytic bacterial killing assisted by nanozyme [159,160], vaccine method, and monoclonal antibody method, etc. These new mechanisms against DRB point out the alternative research direction for the fight against DRB in the “post-antibiotic era”.

## Figures and Tables

**Figure 1 antibiotics-11-01215-f001:**
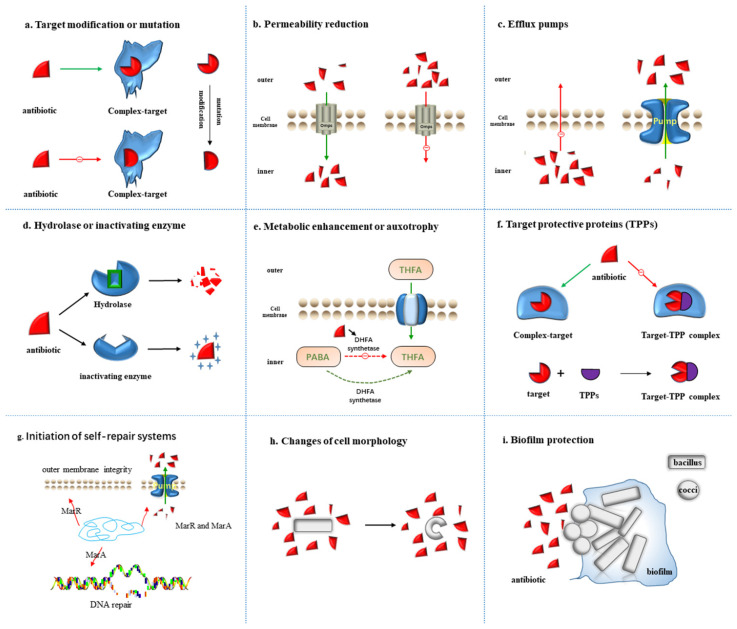
Nine resistance mechanisms of bacteria to antibiotics: (**a**) Target modification or mutation; (**b**) Permeability reduction; (**c**) Efflux pumps; (**d**) Hydrolase or inactivating enzyme; (**e**) Metabolic enhancement or auxotrophy; (**f**) Target protective protein; (**g**) Initiation of self-repair systems; (**h**) Changes of cell morphology; (**i**) Community cooperative resistance. The red triangle indicates antibiotics, and the resistance mechanisms of each figure are described in detail in the main text.

**Figure 2 antibiotics-11-01215-f002:**
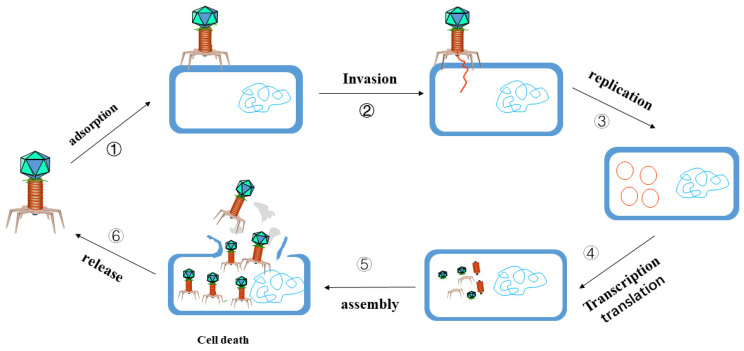
The phage infection process refers to the following six steps [108]: ① adsorption: the process of specific binding between protein attached by phage and host surface receptor protein. Different phages match different cell receptors, which exist on the cell wall, capsule, flagella or sexual pili. ② Invasion: when finding the host, bacteriophage will pierce the surface of the bacteria using its tail filament, and then inject its genetic information into the bacteria through the tail, with the capsid left outside the cell [109]; ③ Synthesis of bacteriophage macromolecules: The nucleic acid and protein of bacteriophage using the phage nucleic acid entering the host as a template; ④ Transcription and translation: depend on the transcription and translation system of bacteria to synthesize substances; ⑤ Assembly: the assembly of phage is manipulated by genetic information, with each part completed in the host according to a certain process; ⑥ Release: most phages are released to the outside of the cell by splitting the cell. In this process, lysozyme hydrolyzes the cell wall, lipase hydrolyzes the cell membrane, so that the cell is cracked and releases the offspring phages.

**Figure 3 antibiotics-11-01215-f003:**
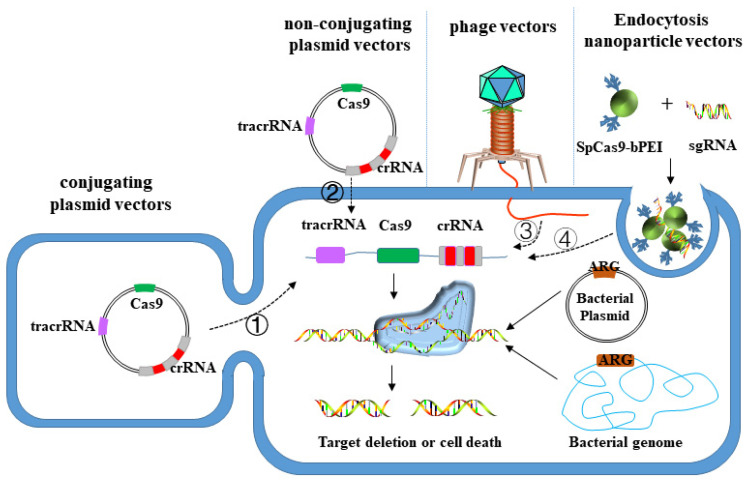
Schematic diagram of CRISPER-Cas editing bacterial drug resistance genes using three different vectors: ① plasmid vector (conjugated transfer); ② plasmid vector (non-conjugative transfer); ③ Phage vector; ④ Nanoparticle vector; different vector-mediated sequences have the same action process in bacterial cells.

**Table 1 antibiotics-11-01215-t001:** The information of the bacteriostatic compound molecules reported in recent years.

Compound Name	Bacteriostatic Spectrum	Action Target	Bacteriostatic Mechanism	Report
Teixobactin	Gram-positive bacteria	cell wall	inhibit cell wall synthesis by binding to a highly conserved motif of lipid II and lipid III	Kim Lewis 2015, [16,93]
Pseudouridimycin	*S. aureus*, etc	RNA polymerase	nucleoside triphosphate to RNA polymerase by occupying the binding site of NTP.	Sonia I Maffioli 2017, [94]
G907	*E. coli*, etc	ATP-binding cassette transporter	inhibit *E. coli* MsbA physiological functions	Christopher M. Koth 2018, [95]
Arylomycin(G0775)	ESKAPE, etc	type I signal peptidase	inhibit the activity of type I signal peptidase	Christopher Heise, 2018, [18]
Chimeric peptidomimetic	Gram-negative bacteria	cell membranes	bind to both lipopolysaccharide and the main component (BamA) of the β-barrel folding complex (BAM)	John A. Robinson 2019, [17]
Darobactin	Gram-negative bacteria	cell membranes	bind to the key outer membrane protein BamA, disrupts the bacterial outer membrane and induces cell lysis	Kim Lewis 2019, [96]
Complestatin and Corbomycin	Gram-positive bacteria	cell wall	block the effect of cell autolysin on cell wall and preventing the collapse of cell wall	Wright, Gerard D 2020, [19]
Halicin	*E. coli*, *M. tuberculosis*, *A. baumannii*, etc	cell membranes	destroy their ability to maintain electrochemical gradients on cell membranes	James J. Collins 2020, [97]
SCH-79797	broad spectrum	folate metabolism and cell membrane	simultaneously targeting folate metabolism and membrane integrity	Zemer Gitai 2020, [98]
Macolacin	Gram-positive Bacteria(*mrc-1*)	cell membranes	a homologue with different structure from colistin, and its antibacterial mechanism is similar to colistin	Sean F. Brady, 2021, [99]
bCSE inhibitors (NL1, NL2, NL3)	*S. aureus,* *P. aeruginosa*	H_2_S synthesize metabolism	inhibit the production of bacterial H_2_S to enhance the bactericidal efficacy of antibiotics	Nudler 2021, [100]
Iboxamycin (IBX)	broad spectrum	ribosome	shift methylated ribosomal nucleotides and expose drug binding sites	Andrew G. Myers, 2021, [20]
Menaquinone-binding antibiotic (MBA)	MRSA, etc	menaquinones	target menaquinones that play a key role in the electronic transmission of bacteria	Sean F. Brady 2021, [101]
Cilagicin	Gram-positive bacteria	cell walls	simultaneous binding of two molecules c55-p and c55-pp that maintain bacterial cell walls	Sean F. Brady 2022, [102]

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
