# Peer review of "The Mechanism of Bacterial Resistance and Potential Bacteriostatic Strategies"

_antibiotics, 2022, doi:10.3390/antibiotics11091215_

Round 1

Reviewer 1 Report

I have some major comments:

Line 52: Please mention carbapenem resistant strains.

carbapenem resistant Acinetobacter baumannii, carbapenem resistant Pseudomonas aeruginosa, and carbapenem resistant and ESBL-producing Enterobacteriaceae (including KlebsiellaE. coliSerratia, and Proteus)

Line 89: staphylococci. spp should be replace with “Staphylococci. spp”. The first letter is capitalized.

Line 134: The name of the bacteria should be written in italics. “S. aureus”

Please check that the name of the bacteria is written in italics in all the text.

Line 135: The gene should be reported in italics. Please check that the names of the genes are written in italics in all the text

Line 136: gene encoding PBP2a, called mecA should be replace with “mecA”. The gene should be reported in italics.

Line 141: It is suggested to use the Gram-negative bacteria abbreviation form in the text. Gram-negative bacteria (GNB), as well as drug-resistant bacteria in the text.

Line 208: It is suggested to briefly explain the most important β-lactamase enzymes (for example ESBLs and carbapenemases) in the 2.4. section. (2.4. Hydrolase or inactivating enzyme).

Line 414: Please write the names of the bacteria in the brackets. The ESKAPE pathogens (Enterococcus faecium, Staphylococcus aureus, Klebsiella pneumoniae, Acinetobacter baumannii, Pseudomonas aeruginosa, and Enterobacter species).

Line 473: Gram-negative pathogens carrying MCR-1 resistance gene should be replace with “Gram-negative pathogens carrying mcr-1 resistance gene”. The gene should be reported in italics. Please check that the names of the genes are written in italics in all the text.

Author Response

Response to Reviewer 1 Comments

Point 1. Line 52: Please mention carbapenem resistant strains.

carbapenem resistant Acinetobacter baumannii, carbapenem resistant Pseudomonas aeruginosa, and carbapenem resistant and ESBL-producing Enterobacteriaceae (including KlebsiellaE. coliSerratia, and Proteus)

Response 1: Your suggestion is very meaningful, which will make the information description of drug-resistant bacteria more accurate and specific. I have revised it in the text according to the suggestion of reviewers.

Point 2. Line 89: staphylococci. spp should be replace with “Staphylococci. spp”. The first letter is capitalized.

Response 2: I am very grateful to the reviewers for their serious and meticulous review, and found that we have problems in many details. Such problems are very inappropriate. I have revised the error and re-existed in the full text, similar problem has been comprehensively checked and revised, thank you again for your rigorous and patient review.

Point 3. Line 134: The name of the bacteria should be written in italics. “S. aureus”. Please check that the name of the bacteria is written in italics in all the text.

Response 3: Thank you for pointing out the errors in the writing format. We have made a comprehensive correction to this problem. Thank you very much for your careful review.

Point 4. Line 135: The gene should be reported in italics. Please check that the names of the genes are written in italics in all the text.

Response 4: Thank you for your valuable suggestions. We have comprehensively revised such questions. Your suggestions have made our paper format more standardized and accurate.

Point 5. Line 136: gene encoding PBP2a, called mecA should be replace with “mecA”. The gene should be reported in italics.

Response 5: I am really sorry for such a problem. We have carefully checked the entire manuscript and have made a comprehensive revision to the similar formatting problems. I would like to express my heartfelt thanks for your suggestions.

Point 6. Line 141: It is suggested to use the Gram-negative bacteria abbreviation form in the text. Gram-negative bacteria (GNB), as well as drug-resistant bacteria in the text.

Response 6: Thank you very much for your suggestion, which will make our manuscript more concise. According to your suggestion, we have revised all the proprietary terms that can be expressed in the form of abbreviations in the main text.

Point 7. Line 208: It is suggested to briefly explain the most important β-lactamase enzymes (for example ESBLs and carbapenemases) in the 2.4. section. (2.4. Hydrolase or inactivating enzyme).

Response 7: Your suggestion is very meaningful, and it will make this part of the manuscript more detailed and perfect. We have supplemented the relevant contents of ESBLs and carbapenemases in this section of the text 2.4 according to your suggestion, and the relevant modified contents can be viewed in the manuscript, so we will not add them here.

Point 8. Line 414: Please write the names of the bacteria in the brackets. The ESKAPE pathogens (Enterococcus faecium, Staphylococcus aureus, Klebsiella pneumoniae, Acinetobacter baumannii, Pseudomonas aeruginosa, and Enterobacter species).

Response 8: Thank you for your suggestion, please forgive our negligence, we have supplemented the explanation of ESKAPE pathogens in the main text, and also checked other similar problems in the full text and made revisions, revisions made to the manuscript have been marked up using the "Track Changes" function.

Point 9. Line 473: Gram-negative pathogens carrying MCR-1 resistance gene should be replace with “Gram-negative pathogens carrying mcr-1 resistance gene”. The gene should be reported in italics. Please check that the names of the genes are written in italics in all the text.

Response 9: I am really sorry for such a problem. We have carefully checked the entire manuscript and have made a comprehensive revision to the existing similar formatting problems. I would like to express my heartfelt thanks again for your suggestion.

Reviewer 2 Report

The authors have done an appreciable job of summarizing the vast topic of drug resistance which is an important issue. I have one general comment and a few specific comments on the manuscript submitted which are as follows:

1) In many cases, authors have referred to other reviews in the reference which is fine. Still, I would suggest authors to add references for original research articles and reviews wherever applicable.

2) I would suggest the author to add on Mtb drug resistance leading due to mutation in rpoB gene sequences causing the drug resistance to frontline anti-TB drugs in section 2.1.

3) To this reviewer's understanding, dormancy may not be considered a mechanism of drug resistance but rather is a mechanism of drug tolerance. Authors should either remove it from the section on mechanisms of drug resistance or provide more reports supporting the author's view of dormancy as a drug resistance mechanism. In fact, the references referred to in the current section also suggest that it is a drug tolerance mechanism and not drug resistance.

4) Authors can add a section on anti-microbial peptides as potential therapeutic approaches, phage therapy, and others listed in the review.

Author Response

Response to Reviewer 2 Comments

Point 1: In many cases, authors have referred to other reviews in the reference which is fine. Still, I would suggest authors to add references for original research articles and reviews wherever applicable. Thank you very much for your very correct and meaningful suggestions. In the citation of references, we try to cite original research papers as much as possible for research questions, and we also use reviews as much as possible for some opinion conclusions.

Response 1: Thank you very much for your very correct and meaningful suggestions. In the citation of references, we try to cite original research papers as much as possible for research questions, and we also use reviews as much as possible for some opinion conclusions. We hope that this will help readers find comprehensive articles in relevant fields quickly. We have also checked and improved the citation of references to make the citation of the paper more accurate. Thank you again for this constructive suggestion.

Point 2: I would suggest the author to add on Mtb drug resistance leading due to mutation in rpoB gene sequences causing the drug resistance to frontline anti-TB drugs in section 2.1.

Response 2: Thank you very much for your very professional suggestion. The drug resistance of M. tuberculosis caused by mutation of rpoB gene is a typical example of bacterial drug resistance caused by gene mutation. According to your suggestion, we have supplemented this part in manuscript 2.1. Target Modification or Mutation, and the relevant modified contents can be viewed in the manuscript, so we will not add them here.

Point 3: To this reviewer's understanding, dormancy may not be considered a mechanism of drug resistance but rather is a mechanism of drug tolerance. Authors should either remove it from the section on mechanisms of drug resistance or provide more reports supporting the author's view of dormancy as a drug resistance mechanism. In fact, the references referred to in the current section also suggest that it is a drug tolerance mechanism and not drug resistance.

Response 3: This view of the reviewer is very correct. We have consulted a large number of related literatures and can only prove that the bacterial dormancy mechanism is a tolerance mechanism rather than a drug resistance mechanism. Therefore, in order to make the paper more rigorous, we decided to delete the content of Section 2.9 according to the reviewer's opinion, and modified the parts related to this section in the paper. Thanks again for the reviewer's rigorous attitude and careful review.

Point 4: Authors can add a section on anti-microbial peptides as potential therapeutic approaches, phage therapy, and others listed in the review.

Response 4: Thanks to the reviewer for this suggestion, antimicrobial peptides are similar to the potential bacteriostatic compounds we introduced in 3.1 Newly potential bacteriostatic compound molecule, so they will not be introduced separately, and phage therapy has also been highlighted in Section 3.2. Nanomaterials for antimicrobial-free antimicrobial application, vaccine method and monoclonal antibody method, which are mentioned in the manuscript, have great research value and application prospect, but this review focuses on mechanism of bacterial resistance and three antimicrobial methods introduced in the manuscript, so other antimicrobial methods are not summarized.

Reviewer 3 Report

Zhang et al have presented a comprehensive review on the mechanisms of antibiotic resistance and highlighted the most recent strategies of combating antibiotic resistance.

The authors elaborate on each of the possible mechanisms of drug resistance in bacteria and discuss alternative therapy options with Phage therapy and CRISPR-Cas9 mediated modification of target proteins.

The authors have presented a detailed review of literature. I would however recommend that the authors proof read the review for better clarity. I have a few additional comments which would be helpful for the authors to address.

1.     In the abstract, the authors claim 3 kinds of potential antibacterial methods but only state 2. Kindly clarify (Line 24)

2.     Explain abbreviation BlaNDM-1: metallo-β-lactamase gene. Provide appropriate citation (Yong et al) . It was identified first in 2009 in Klebsiella pneumoniae. (Line 47-48)

Yong D. et al. Characterization of a new metallo-β-lactamase gene, blaNDM-1 and a novel erythromycin esterase gene carried on a unique genetic structure in Klebsiella pneumoniae sequence type 14 from India. Antimicrob Agents Chemother 53, 5046–5054 (2009).

3.     Explain briefly the mechanism of action of each drug or perhaps what pathway/protein it is targeting ( for eg : type 1 signal peptidase, ribsomes etc)

4.     Explain figure 1 legend in more detail. Alternatively provide details of mechanism in text in the case of fig 1e.

5.     Please provide citation: for lines 310-319

6.     Kindly clarify “play release the efficacy” in line 321. N

7.     Line 326, The citation describes the WT form of HipA that phosphorylates GltX and not the mutant form. 

Author Response

Response to Reviewer 3 Comments

Point 1: In the abstract, the authors claim 3 kinds of potential antibacterial methods but only state 2. Kindly clarify (Line 24).

Response 1: Thank you very much for pointing out this problem for us. The description of the three antibacterial methods mentioned in the abstract is not clear enough. In order to make readers understand more clearly, we have rewritten this description, and the specific modification forms can be checked in the manuscript.

Point 2: Explain abbreviation BlaNDM-1: metallo-β-lactamase gene. Provide appropriate citation (Yong et al). It was identified first in 2009 in Klebsiella pneumoniae(Line 47-48)

Yong D. et al. Characterization of a new metallo-β-lactamase gene, blaNDM-1 and a novel erythromycin esterase gene carried on a unique genetic structure in Klebsiella pneumoniae sequence type 14 from India. Antimicrob Agents Chemother 53, 5046–5054 (2009).

Response 2: Thank you for your meticulous and rigorous advice. This is an error caused by our negligence in consulting literature. We have revised this issue in the text. We also checked the abbreviations that first appeared in the manuscript and explained them where appropriate.

Point 3:  Explain briefly the mechanism of action of each drug or perhaps what pathway/protein it is targeting (for eg: type 1 signal peptidase, ribsomes etc).

Response 3: Thank the reviewers for their very professional suggestions. In section 3.1, the antibacterial mechanism of various compounds is described and summarized in table1. However, there is no specific introduction about the target of each compound. Therefore, we have supplemented the target of each compound in Table 1. Please refer to the revised Table 1 in the manuscript for the revised content.

Point 4:  Explain figure 1 legend in more detail. Alternatively provide details of mechanism in text in the case of fig 1e.

Response 4 Thank you very much for the very meaningful questions pointed out by the reviewers. Due to our negligence, the explanation of fig 1e in the manuscript is not clear enough. Therefore, we have fully explained the drug resistance mechanism described in fig 1e. For the revised content, please refer to section 2.5. Metaphysical alteration or auxotrophy of the manuscript.

Point 5: Please provide citation: for lines 310-319.

Response 5: I'm very sorry for the negligence of our work. The following reference can demonstrate this view, and we have quoted it in the manuscript. Thanks to the reviewers for their meticulous suggestions.

Reference: Mickiewicz, K.M.; Kawai, Y.; Drage, L.; Gomes, M.C.; Davison, F.; Pickard, R.; Hall, J.; Mostowy, S.; Aldridge, P.D.; Errington, J. Possible role of L-form switching in recurrent urinary tract infection. Nature Communications 2019, 10, 4379, doi:10.1038/s41467-019-12359-3.

Point 6:  Kindly clarify “play release the efficacy” in line 321. N

Response 6: In view of the fact that we have deleted the content in Section 2.9 of the original manuscript, we will not make redundant description of this issue, but we still thank the reviewers for their valuable suggestions.

 Point 7: Line 326, The citation describes the WT form of HipA that phosphorylates GltX and not the mutant form. 

Response 7 In view of the fact that we have deleted the content in Section 2.9 of the original manuscript, we will not make redundant description of this issue, but we still thank the reviewers for their valuable suggestions.